# Mixed-View Panorama Synthesis using Geospatially Guided Diffusion

**Zhexiao Xiong**                                                                                     *x.zhexiao@wustl.edu*
*Department of Computer Science & Engineering, Washington University in St. Louis*

**Xin Xing**                                                                                           *xxing@unomaha.edu*
*Department of Computer Science, University of Nebraska Omaha*

**Scott Workman**                                                                                 *scott.workman.ai@gmail.com*

**Subash Khanal**                                                                                     *k.subash@wustl.edu*
*Department of Computer Science & Engineering, Washington University in St. Louis*

**Nathan Jacobs**                                                                                     *jacobsn@wustl.edu*
*Department of Computer Science & Engineering, Washington University in St. Louis*

**Reviewed on OpenReview:** *https://openreview.net/forum?id=ylUVRikhTL*

## Abstract

We introduce the task of mixed-view panorama synthesis, where the goal is to synthesize a novel panorama given a small set of input panoramas and a satellite image of the area. This contrasts with previous work which only uses input panoramas (same-view synthesis), or an input satellite image (cross-view synthesis). We argue that the mixed-view setting is the most natural to support panorama synthesis for arbitrary locations worldwide. A critical challenge is that the spatial coverage of panoramas is uneven, with few panoramas available in many regions of the world. We introduce an approach that utilizes diffusion-based modeling and an attention-based architecture for extracting information from all available input imagery. Experimental results demonstrate the effectiveness of our proposed method. In particular, our model can handle scenarios when the available panoramas are sparse or far from the location of the panorama we are attempting to synthesize. The project page is available at `https://mixed-view.github.io`.

## 1 Introduction

The wide availability of street-level panoramas and their integration into mapping applications has dramatically improved the navigation experience for users. Access to nearby panoramas reduces the difficulty of navigating from a purely overhead map. However, an inherent problem is that panoramas are expensive to collect and thus are sparsely available and updated infrequently, with many roads having no panoramas. This has motivated the task of cross-view synthesis Zhai et al. (2017); Regmi & Borji (2018); Shi et al. (2022a), where street-level panoramas are synthesized directly from satellite imagery. Unfortunately, existing approaches ignore any available nearby street-level panoramas at inference time.

We propose a new task, mixed-view panorama synthesis (MVPS), which also aims to synthesize a street-level panorama. However, available street-level panoramas are also used to control the synthesis process. Figure. 1 gives a visual overview of the proposed mixed-view setting. In contrast to many recent works on novel view synthesis of outdoor scenes (e.g., NeRF-related works (Mildenhall et al., 2021; Martin-Brualla et al., 2021; Xie et al., 2023; Tancik et al., 2022) and 3DGS-related works (Kerbl et al., 2023; Zhou et al., 2024; Liu et al., 2024)), which require dense images with accurate camera pose information captured under

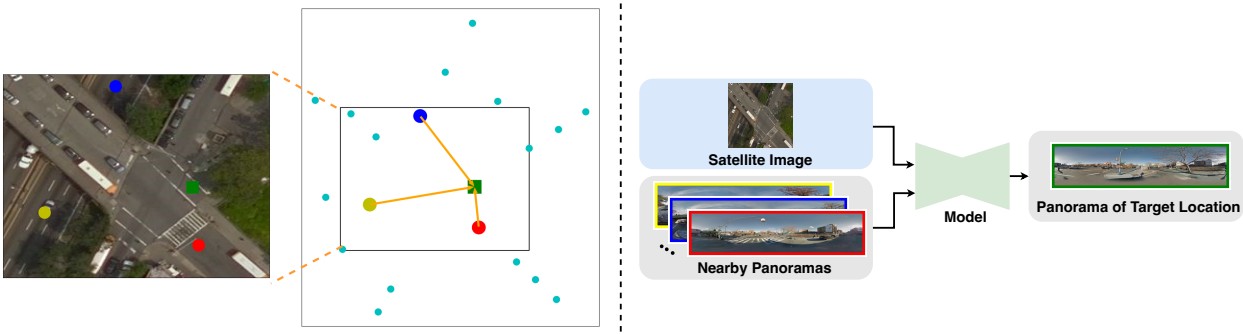

Figure 1: We propose a new task, mixed-view panorama synthesis, in which a satellite image and a set of nearby panoramas (blue, yellow, and green) are used to render a panorama at a novel location (green). Our approach uses diffusion-based modeling and attention to enable flexible, multimodal control.

fairly controlled settings, our input data are sparsely distributed panoramas, with only geo-location and orientation information provided. In particular, the distance, orientation, and availability of nearby street-level panoramas can vary dramatically. This task is in many ways more challenging than the text-to-image generation problem. First, previous text-to-image generation methods only focus on semantic accuracy, while in our task, geometric faithfulness is the primary factor to be considered. Second, nearby panoramas are often not acquired simultaneously, leading to significant challenges with non-stationary objects, lighting variation, and seasonal changes. Therefore, a method needs to be able to condition the output on images from different viewpoints, both street-level and overhead, using the geometric relationship between them.

Recently, conditional diffusion models Rombach et al. (2022) have been widely used in the image synthesis task. However, most of them are typically restricted to single-condition inputs, which is unsuitable for our MVPS task that aims to use multiple conditions and their correspondence to synthesize complex outdoor scenes. ControlNet Zhang et al. (2023) extends the pretrained Stable Diffusion model to take multi-modal data as inputs, while ControlNet-based approaches still focus primarily on conditional inputs that are spatially aligned with the target image, such as sketches, depth maps, and segmentation maps, which are essentially different forms of representation of the target image. In addition, recent works on attention-manipulated diffusion models predominantly focus on manipulating the text modality via cross-attention module. For outdoor panorama synthesis, with complex scene layouts, text alone is not sufficient to achieve fine-grained spatial control. Limitations of current diffusion frameworks motivate the need for frameworks that allow multi-conditioned and fine-grained geometry control for outdoor panorama synthesis. To address these challenges, there is a pressing need to develop a framework capable of handling multiple image input controls from diverse modalities, with the ability to guide the model to focus on the regions of the input images with salient geometric correspondence to the target image.

In this work, we propose a multi-conditioned, end-to-end diffusion framework for combining information from all input imagery to guide the diffusion-based mixed-view synthesis process. A key element of our approach is the integration of geospatial attention Workman et al. (2022) to guide the controllable diffusion model for MVPS task. Geospatial attention incorporates the semantic content of the input, geometry, and overhead appearance to identify the geo-informative regions of an input panorama relative to the target location. We extend the concept of geospatial attention to include both local-level and global-level attention. The resulting attention maps are fused with the corresponding input images in the latent space of the diffusion model. Finally, the attention-guided features are integrated into the encoders of the corresponding conditional branch in the multi-conditioned ControlNet model, achieving geometric-guided, fine-grained spatial control.

The key contributions of this work can be summarized as:

- We propose the mixed-view panorama synthesis task: using a satellite image and a sparse set of nearby panoramas to synthesize a target panorama at a given location.

- We propose a unified multi-conditioned controllable diffusion framework, GeoDiffusion, for the mixed-view panorama synthesis task.

- We use geospatial attention as the geometric constraint to associate nearby views with the target view to achieve geometry-guided fine-grained spatial control.

- We validate that our proposed method generates high-fidelity panorama images with geometric accuracy on the Brooklyn and Queens benchmark dataset, achieving state-of-the-art performance while having more flexibility compared with prior cross-view synthesis works.

## 2 Related Work

### 2.1 Cross-view Synthesis

Given a satellite image, the cross-view synthesis task aims to predict the street-level panorama. Prior work includes using a learned linear transformation Zhai et al. (2017), applying conditional GANs Regmi & Borji (2018; 2019); Tang et al. (2019); Li et al. (2021), integrating height estimation as explicit supervision Shi et al. (2022a), using a density field representation Qian et al. (2023) and point-based neural rendering Li et al. (2024). Structure-preserving panorama synthesis methods have also shown to be effective in the related task of cross-view image geolocalization Regmi & Shah (2019); Toker et al. (2021); Shi et al. (2022b). However, such methods typically only consider a satellite image as input and only attempt to synthesize street-level panoramas in the setting where the satellite image is center-aligned with the target location. Our work integrates near and remote modalities (i.e., mixed-view) and can synthesize panoramas at arbitrary locations in the given satellite image region.

### 2.2 Image-to-Image Translation

The goal of image-to-image translation is to learn a mapping between an input image and a target image Isola et al. (2017). Traditional methods are based on generative adversarial networks Park et al. (2019); Wang et al. (2018a); Liu et al. (2017); Zhu et al. (2017). With the development of vision transformers, several works have successfully applied transformers to this problem Esser et al. (2021); Chang et al. (2022). More recently, numerous methods have leveraged diffusion model to perform image translation Dhariwal & Nichol (2021); Rombach et al. (2022); Xue et al. (2023); Parmar et al. (2023); Wang et al. (2022). However, these methods are unable to handle cases in which direct conditions are unknown and only implicit geometric relationships are available. In contrast, our model is capable of handling multiple indirect image controls and utilizing the geometric relationships both within and across different input modalities.

### 2.3 Conditional Diffusion Models

Conditional diffusion models enable controllable image synthesis and editing. Most recent work in this domain focuses on the text-to-image synthesis problem, such as GLIDE Nichol et al. (2021), DALL-E2 Ramesh et al. (2022) and Stable Diffusion Rombach et al. (2022). These methods require large training datasets containing many image-text pairs to generate high-quality images. However, generating complex scenes is challenging with only text information. Recent works have proposed incorporating local, spatial conditioning, such as segmentation maps Mou et al. (2024) or layouts Li et al. (2023); Zheng et al. (2023); Qu et al. (2023), to overcome these challenges and achieve precise spatial control. ControlNet Zhang et al. (2023) and similar methods Qin et al. (2023); Zhao et al. (2023) use zero-convolution layers to incorporate task-specific conditioning into pretrained image diffusion models, significantly reducing the computational cost and sample complexity while still generating high-quality images. We adopt this approach to build a multi-conditioned, end-to-end geospatial attention-guided diffusion framework.

### 2.4 Attention

Attention mechanisms have shown benefits in a variety of visual tasks. Commonly used attention mechanisms include channel-wise attention Hu et al. (2018b); Woo et al. (2018), spatial attention Mnih et al. (2014);

Jaderberg et al. (2015); Hu et al. (2018a); Wang et al. (2018b), spatial-temporal attention Meng et al. (2019); Song et al. (2017), and branch attention Fukui et al. (2019). Self-attention Vaswani et al. (2017) and cross-attention Chen et al. (2021) are widely applied in vision transformers Dosovitskiy et al. (2020); Carion et al. (2020); Xie et al. (2021). Recently latent diffusion models Rombach et al. (2022); Wu et al. (2023); Xue et al. (2023), also regard cross-attention as an effective way to allow multi-modal training for class-conditional, text-to-image, and spatially conditioned tasks. Besides cross-attention, geometry has been used to inform the learning of attention, such as using epipolar attention He et al. (2020) in novel view synthesis Tseng et al. (2023). Workman *et al.* Workman et al. (2022) proposed the concept of geospatial attention, a geometry-aware attention mechanism that explicitly considers the geospatial relationship between the pixels in a ground-level image and a geographic location. We extend this concept to our mixed-view panorama synthesis task to identify geo-informative regions across mixed viewpoints, allowing our diffusion model to be 'geometry aware.'

## 3 Preliminaries

### 3.1 Stable Diffusion Architecture

Stable Diffusion Rombach et al. (2022) is a generative modeling approach via a learned diffusion process, by applying the diffusion operation in the latent space. It uses a UNet-like structure as its denoising model, which consists of an encoder, a middle block, and a decoder. Both the encoder and the decoder are made up of 12 blocks, and the full model contains 25 blocks. The outputs of the encoder are added to the 12 skip-connections and 1 middle block of the UNet. The input for the $i$-th block in the decoder is represented as:

$$\begin{cases} \text{concat}\,(m, f_j) & \text{where } i = 1, j = 13 - i \\ \text{concat}\,(g_{i-1}, f_j) & \text{where } 2 \leq i \leq 12, j = 13 - i \end{cases} \tag{1}$$

where $m$ denotes the output of the middle block and $f_i$ and $g_i$ denote the output of the $i$-th block in the encoder and decoder, respectively.

### 3.2 Mixed-View Panorama Synthesis

We introduce the task of mixed-view panorama synthesis (MVPS). Given a satellite image $S_1$, and a set of sparsely distributed nearby street-level panoramas $(P_1, P_2, \cdots, P_n)$ with known geolocations $(l_1, l_2, \cdots, l_n)$, we aim to synthesize a panorama $P_t$ in the region of $S_1$ at a target location $l_t$. To generate the target panorama $P_t$ with precise layout distribution, the synthesis process should utilize: (1) the semantic information from $P_1, P_2, \cdots, P_n$ and $S_1$, (2) the geometric relationships between street-level $P_i \rightarrow P_t, i \in (1, 2, \cdots, n)$ and across satellite & street-level $S_1 \rightarrow P_t$. With only location information provided, how to utilize the implicit geometric relationships is the key challenge of this task.

## 4 Geospatial Attention-Guided Diffusion Model

We propose a novel multi-conditioned, geospatial attention-guided diffusion model, GeoDiffusion, which combines information from a satellite image and nearby street-level panoramas to synthesize a target panorama. Figure. 2 shows an overview of our framework, which consists of two main modules: (1) a novel geospatial attention adapter that combines information from the input imagery and (2) a multi-conditioned diffusion model, based on ControlNet Zhang et al. (2023), that synthesizes the target panorama.

### 4.1 Geospatial Attention Adapter

We propose an adapter that uses geospatial attention Workman et al. (2022) to fuse image features that are extracted using CNN-based encoders. The geospatial attention adapter has two components: (1) local attention for assigning weight to panorama features and (2) global attention for assigning weight to satellite features. These attention maps are used to fuse features from the CNN-derived feature maps, which subsequently control the diffusion model.

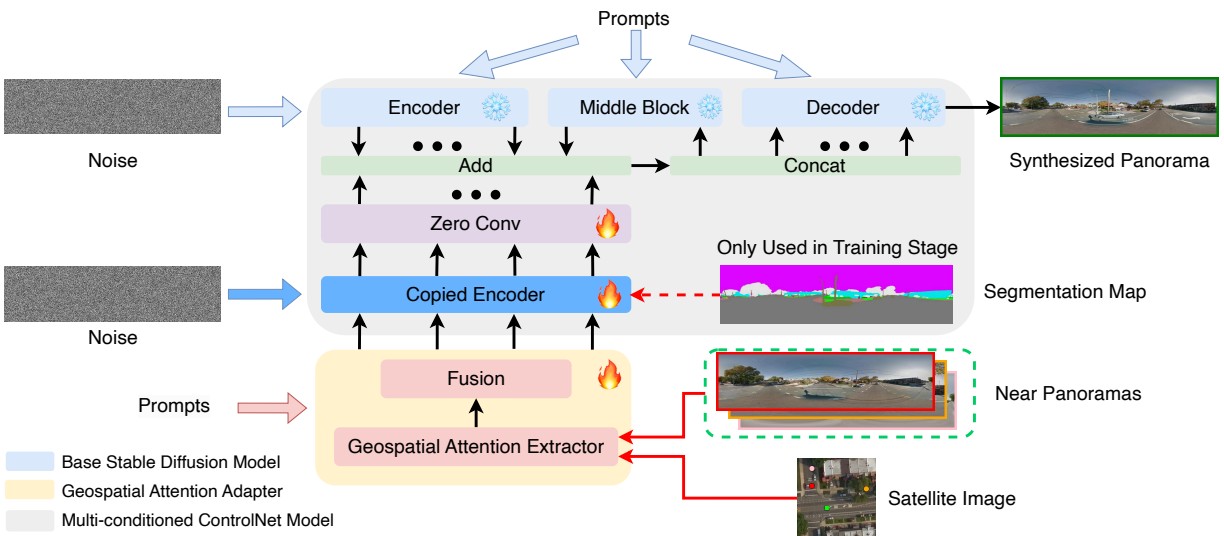

Figure 2: The overall framework of our model. For each branch of the near panoramas in the multi-conditioned ControlNet diffusion model, the near panoramas and the satellite image are passed through the geospatial attention adapter and the copied encoder, and the extracted features are injected into the Stable Diffusion encoder.

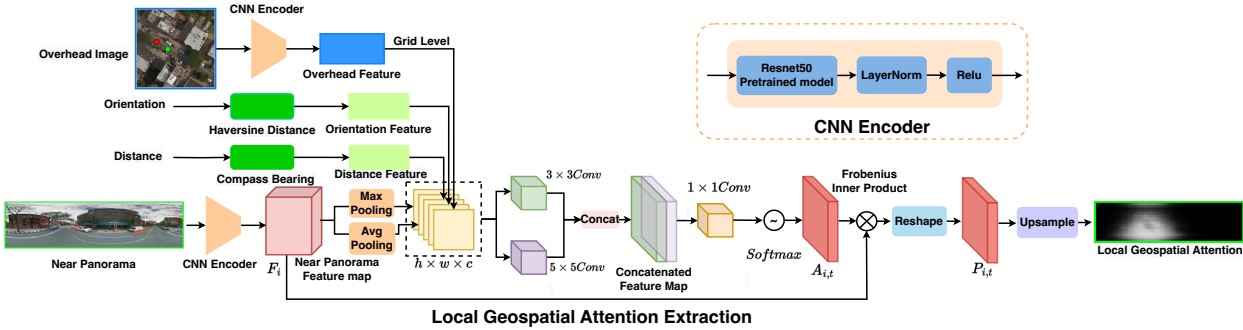

Figure 3: The local geospatial attention extraction module, which provides attention for near panoramas. The red dot represents the target location, and the green dot represents the source panorama location in the overhead image region.

**Extracting Local Attention** To identify geoinformative regions in a nearby street-level panorama, we build a local geospatial attention module, shown visually in Figure. 3. In our setting, we assume images are fully calibrated. Given a panorama at location $l_i$ and the target location $l_t$, for MVPS, we apply geospatial attention Workman et al. (2022) based on the relative location, the target-relative orientation of each pixel ray, the semantic content of the input, and the overhead appearance, and get a spatial attention map, $A_{i,t} \in R^{H \times W}$. Based on $A_{i,t}$ and the feature map of the near panorama $F_i \in R^{H \times W \times C}$, we compute an attention-weighted feature descriptor $P_{i,t}$ as equation 2:

$$P_{i,t} = reshape(\langle \mathbf{f}^c, A_{i,t} \rangle_F) \tag{2}$$

where $\mathbf{f}^c \in R^{H \times W}$ represents the input feature map of the $c$-th channel of $F_i$, $\langle ., . \rangle_F$ denotes the Frobenius inner product of the two inputs, and $P_{i,t}$ represents the feature output, which represents the information from the input feature map $F_i$ that is relevant to the target location $l_t$. We detail the extraction of local-level geospatial attention in the supplementary material.

**Extracting Global Attention** One of the key challenges in MVPS is handling the nonuniform spatial distribution of the input panoramas. To address this, we extract the spatial distributions of the panorama

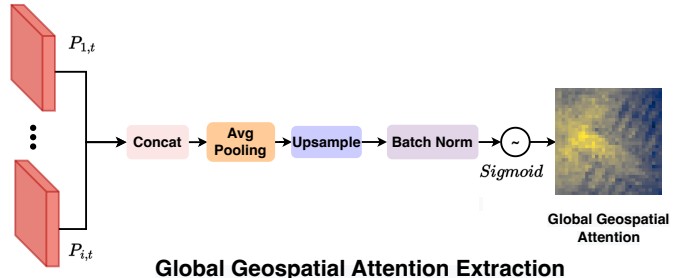

Figure 4: Global geospatial attention extraction module, which provides attention for satellite images.

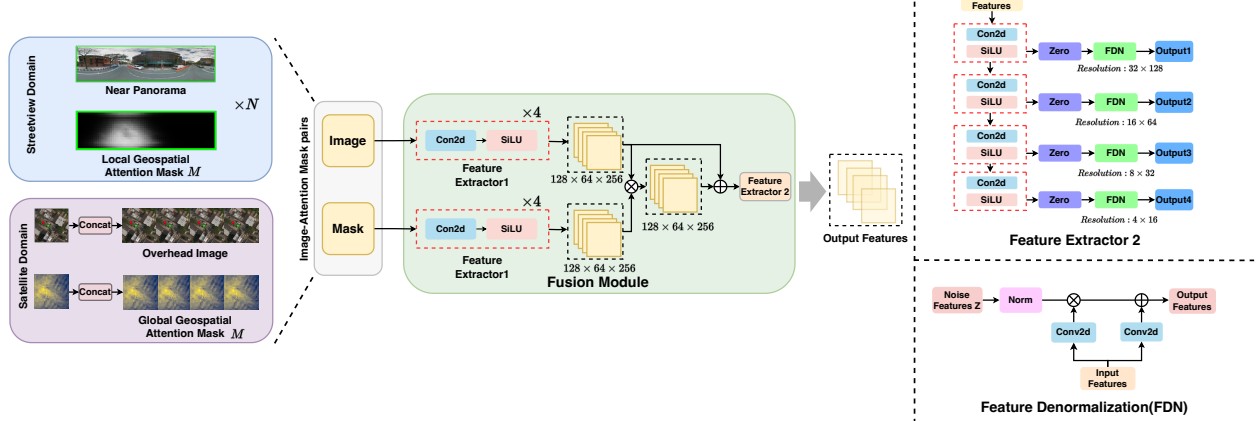

Figure 5: The geospatial attention fusion module. We use the same architecture to add attention to near panoramas and satellite image respectively in the latent space, and inject the features into the copied encoder of the diffusion model.

features in the given region and propose global geospatial attention, which is an attention map that is used to guide which features should be more heavily weighted in the feature space of satellite image. See Figure. 4 for an overview. Given a target location $l_t$, this module operates on the attention-weighted features for the input street-level panoramas computed using local-level geospatial attention $P_{1,t}, P_{2,t}, \cdots, P_{i,t}$, where $i$ is the number of panoramas. We aggregate those features using concatenation and average pooling, and get a $32 \times 32$ weight feature grid. We then upsample this dense grid to $256 \times 256$ and pass it through a batch normalization layer and a sigmoid layer to obtain the final global-level geospatial attention. The resulting attention map is the global-level feature distribution in the overhead-view depending on the relative location of the target panorama and the input panoramas.

**Fusion** After extracting the attention mask from both street-level and satellite views, we fuse them with the corresponding street-level panoramas and the satellite image. The fusion module is shown in the left part of Figure. 5. For the street-level panoramas, we use a feature extractor composed of stacked convolutions to extract the attention mask and the near panoramas to the latent space. Both of their shapes are $B \times 128 \times 64 \times 256$ and we fuse them using a Hadamard product. We also add skip connections between the masked near panorama feature map and the original near panorama feature map, which is shown as:

$$H_{i,c}(x) = (1 + M_{i,c}(x)) \odot F_{i,c}(x) \tag{3}$$

where $M(x)$ is the latent space geospatial attention mask ranging from $[0, 1]$, $i$ ranges over all spatial positions, and $c \in \{1, \cdots, C\}$ is the index of the channel. When $M(x)$ approximates 0, the output feature $H(x)$ will approximate original latent features $F(x)$. For the satellite image, based on the extracted satellite attention, we use the same method as nearby street-level panoramas to get the attention-guided feature map. Finally, the output features are passed through a multi-scale feature extractor to get the output features.

We show the detailed structure of the multi-scale feature extractor in the right part of Figure. 5. In the multi-scale feature extractor, for each resolution, we first pass the feature through a zero-convolution layer, which progressively starts influencing the generation with the attention-guided features. Then we adopt Feature Denormalization (FDN) Park et al. (2019) as injection module, which uses the condition features for GeoDiffusion to rectify its normalized input noise features as:

$$FDN_r(Z_r, c_l) = norm(Z_r) \cdot (1 + conv_\gamma(zero(h_r(c_l)))) + conv_\beta(zero(h_r(c_l))), \tag{4}$$

where $Z_r$ is the noise features at resolution $r$, $c_r$ represents the concatenated conditions, $h_r$ represents the output of the feature extractor $H$ at resolution $r$, and $conv_\gamma$ and $conv_\beta$ refer to the learnable convolutional layers that convert condition features into spatial-sensitive scales and shift modulation coefficients respectively. The final extracted features are concatenated and injected into the respective copied encoders. We select the first block of each resolution in the copied encoder for feature injection, corresponding to the $1, 4, 7, 10$ layers of the copied encoder respectively. Leveraging the effectiveness of geospatial attention, the multi-conditioned diffusion network can generate more geometrically accurate panoramas.

## 4.2 Multi-Conditioned Diffusion Model

We introduce our approach based on the conditioning features extracted from the overhead imagery and nearby panoramas. Inspired by ControlNetZhang et al. (2023), we construct a multi-conditioned diffusion model for MVPS (Figure. 2). We start from a pretrained Stable Diffusion Rombach et al. (2022) model and duplicate the structures and weights of the encoder and middle block for each condition $i$, which we refer to as $F_i'$ and $M_i'$ respectively. The conditions are concatenated in the channel dimension, passed through the geospatial attention adapter, and then the attention-guided features for each condition are injected into the corresponding copied encoder, with noise added according to the time embedding. The encoded features are passed through a zero-convolution layer and incorporated into the main branch of the Stable Diffusion architecture. During the decoding process, we keep all other elements unchanged while modifying the input of the $i$-th block of the decoder as:

$$\begin{cases} concat\left(m + m', f_j + zero\left(f_j'\right)\right) & \text{where } i = 1, j = 13 - i \\ concat\left(g_{i-1}, f_j + zero\left(f_j'\right)\right) & \text{where } 2 \leq i \leq 12, j = 13 - i \end{cases} \tag{5}$$

in which $m$ denotes the output of the middle block, $f_i$ and $g_i$ denote the output of the $i$-th block in the encoder and decoder of UNet respectively, and $zero$ represents a zero convolutional layer. Following ControlNet Zhang et al. (2023), the weights of zero convolutional layer is set to increase from zero to gradually add control information into the main Stable Diffusion model. For the text input, as we do not rely on text embedding to achieve content manipulation, we use "A high-resolution street-view panorama" as the default prompt.

## 5 Experiments

We evaluate our approach for mixed-view panorama synthesis quantitatively and qualitatively through various experiments. Results show that our approach, which can take advantage of nearby street-level panoramas, significantly improves results compared to existing cross-view methods, and shows more flexibility in synthesizing the panorama in arbitrary locations.

**Dataset** We train and evaluate our methods using the Brooklyn and Queens dataset Workman et al. (2017). This dataset contains non-overlapping satellite images (approx. 30 cm resolution) and street-level panoramas from New York City collected from Google Street View. It is composed of two subsets collected from Brooklyn and Queens respectively. The Brooklyn subset contains 43,605 satellite images and 139,327 panoramas. The Queens subset, which we use solely for cross-domain evaluation, contains 10,044 satellite images and 38,630 panoramas. For evaluation on the Brooklyn subset, we use the original train/test split, resulting in 38,744 images for training, 500 images for validation, and 4361 images for testing. For cross-domain evaluation on Queens, we randomly select 1000 images from the Queens subset and report the performance on the selected images. During training, when computing geospatial attention, we set the

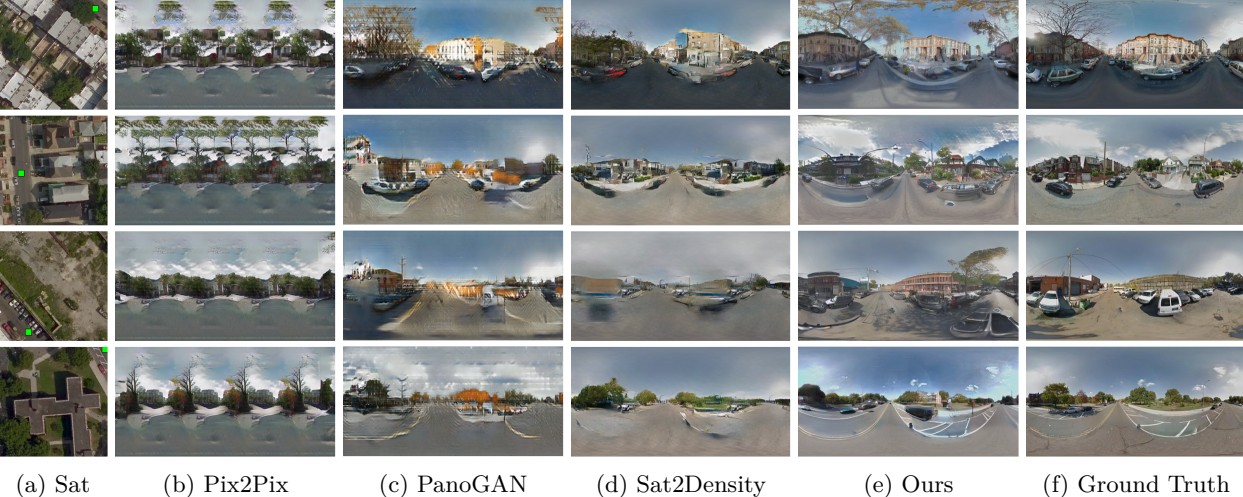

|       |       |       |       |       |       |
|-------|-------|-------|-------|-------|-------|
| (a) Sat | (b) Pix2Pix | (c) PanoGAN | (d) Sat2Density | (e) Ours | (f) Ground Truth |

Figure 6: Qualitative results versus baselines. The cross-view synthesis methods that we compare with are trained on our collected center-aligned satellite images. Our approach, which integrates nearby street-level panoramas, not only generates more realistic results when compared to baselines, but more accurate results both semantically and geometrically when compared to the ground truth.

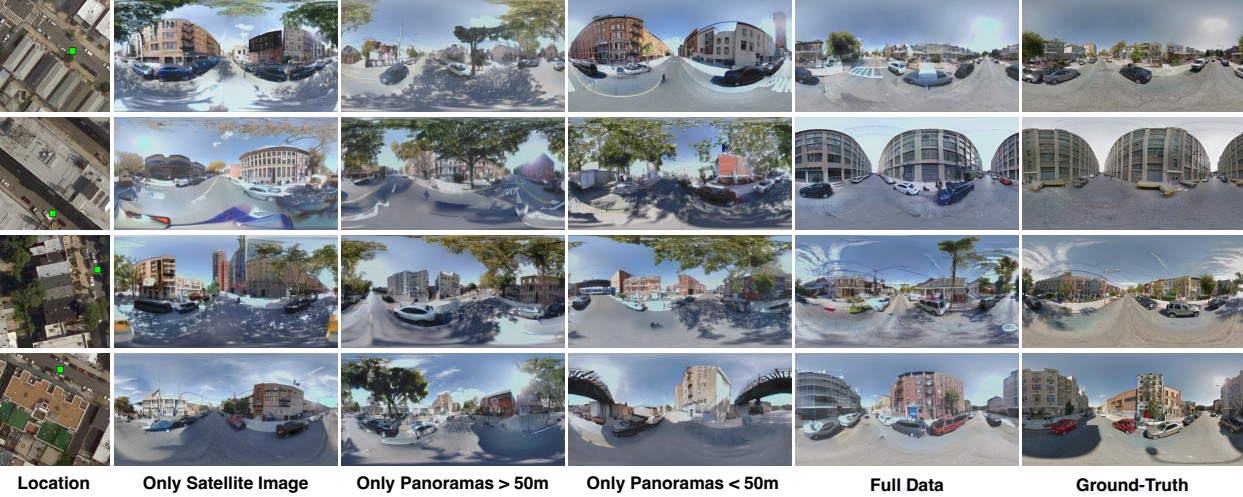

|       |       |       |       |       |       |
|-------|-------|-------|-------|-------|-------|
| **Location** | **Only Satellite Image** | **Only Panoramas > 50m** | **Only Panoramas < 50m** | **Full Data** | **Ground-Truth** |

Figure 7: Qualitative results for MVPS using different sources of data.

number of nearby street-level panoramas considered, $N$, to 20 for each satellite image; In the conditional diffusion module, for each pair of data, we use the 2 closest panoramas to the target location as conditions. During the inference stage, the geospatial attention module is frozen, so only 2 nearby panoramas are in need.

**Metrics** For evaluating the performance of our approach, we consider two classes of metrics. The first class is low-level metrics which evaluate the pixel-wise similarity between two images: peak signal-to-noise ratio (PSNR), structural similarity index (SSIM), root-mean-square error (RMSE), and sharpness difference (SD). The second class is high-level metrics which evaluate image-level differences between two images. For perceptual similarity (LPIPS) Zhang et al. (2018), we use a pretrained AlexNet Krizhevsky et al. (2012) and denote it as $P_{alex}$. We also adopt Fréchet inception distance (FID) Heusel et al. (2017), a common metric for measuring the realism and diversity of images produced by generative models.

Table 1: Comparison with cross-view synthesis methods on Brooklyn test set. Center-Aligned: use satellite data that are center-aligned with the target location.

| Center-Aligned | Method | PSNR↑ | SSIM↑ | $P_{alex}$↓ | RMSE↓ | FID↓ | SD↑ |
|---|---|---|---|---|---|---|---|
| ✗ | Pix2pix Isola et al. (2017) | 11.93 | 0.0950 | 0.6161 | 64.75 | 413.29 | 9.07 |
|  | PanoGAN Wu et al. (2022) | 13.10 | 0.2981 | 0.5583 | 56.98 | 166.30 | 12.04 |
|  | Sat2density Qian et al. (2023) | 13.39 | 0.4325 | 0.5407 | 55.38 | 153.32 | 13.22 |
|  | GeoDiffusion (**Ours**) | **14.14** | **0.4329** | **0.4343** | **51.42** | **33.68** | **13.60** |
| ✓ | Pix2pix Isola et al. (2017) | 12.19 | 0.3375 | 0.5503 | 63.14 | 263.97 | 11.71 |
|  | PanoGAN Wu et al. (2022) | 13.64 | 0.4044 | 0.4856 | 53.71 | 130.98 | 13.28 |
|  | Sat2Density Qian et al. (2023) | 14.57 | 0.4465 | 0.4684 | 48.54 | 87.77 | 13.66 |
|  | GeoDiffusion (**Ours**) | **14.66** | **0.4498** | **0.4206** | **48.31** | **31.07** | **13.79** |

Table 2: Cross dataset evaluation on Queens. Center-Aligned: use satellite data that are center-aligned with the target location.

| Center-Aligned | Method | PSNR↑ | SSIM↑ | $P_{alex}$↓ | RMSE↓ | FID↓ | SD↑ |
|---|---|---|---|---|---|---|---|
| ✗ | Pix2pix Isola et al. (2017) | 11.82 | 0.1719 | 0.6435 | 65.74 | 386.41 | 9.51 |
|  | PanoGAN Wu et al. (2022) | 12.79 | 0.2968 | 0.5770 | 59.24 | 183.82 | 12.02 |
|  | Sat2Density Qian et al. (2023) | 12.93 | 0.4023 | 0.5524 | 58.63 | 181.45 | 13.31 |
|  | GeoDiffusion (**Ours**) | **13.54** | **0.4239** | **0.4661** | **55.80** | **57.66** | **13.55** |
| ✓ | Pix2pix Isola et al. (2017) | 11.77 | 0.2537 | 0.5450 | 66.73 | 265.53 | 10.83 |
|  | PanoGAN Wu et al. (2022) | 13.45 | 0.4014 | 0.4933 | 57.01 | 99.56 | 13.08 |
|  | Sat2Density Qian et al. (2023) | 13.87 | 0.4478 | 0.4835 | 53.76 | 107.14 | 13.61 |
|  | GeoDiffusion (**Ours**) | **14.08** | **0.4488** | **0.4585** | **53.46** | **54.26** | **13.72** |

Table 3: Ablation study for geospatial attention. Local: local-level attention. Global: global-level attention.

| Local | Global | PSNR↑ | SSIM↑ | $P_{alex}$↓ | RMSE↓ | FID↓ | SD↑ |
|---|---|---|---|---|---|---|---|
|  |  | 11.69 | 0.3567 | 0.5089 | 67.41 | 52.56 | 12.60 |
| ✓ |  | 12.95 | 0.3757 | 0.4686 | 58.49 | 37.90 | 12.79 |
|  | ✓ | 12.83 | 0.3898 | 0.4757 | 59.45 | 35.54 | 12.89 |
| ✓ | ✓ | **14.14** | **0.4329** | **0.4343** | **51.42** | **33.68** | **13.60** |

**Implementation Details** We use a pretrained Stable Diffusion Rombach et al. (2022) model (v1.5) with the default parameters. For optimization, we use AdamW with a learning rate of $\lambda = 2 \times 10^{-5}$. The input images are resized to $256 \times 1024$ as local conditions. Specifically, the satellite image is resized to $256 \times 256$ and replicated horizontally four times. We adopt ViT-Adapter Chen et al. (2022) to get the segmentation map of the target panorama. We use DDIM Song et al. (2020) for sampling with the number of time steps set to 50. The classifier free guidance Ho & Salimans (2022) is set to 7.5. During training, the segmentation map of the target image, the satellite image, and a set of nearby street-level panoramas are passed through the controllable diffusion model to synthesize the target panorama. At inference, only the satellite image and available nearby panoramas are required to synthesize the target panorama image, following the same assumption made in Regmi & Borji (2018) since segmentation maps are unlikely to be available in the real world.

**Modality Dropout** Using multi-conditioned inputs results in the high reliance on one or a subset of the conditions. To avoid this, we implement a modality dropout strategy in three forms: 1) randomly omit each individual condition 2) randomly keep all conditions and 3) randomly drop all conditions. This allows the model to learn the mixed-view panorama synthesis task based on arbitrary conditions, reducing the model's reliance on individual conditions, and helping learn the relationships between different modality compositions. In our experiments, during training, we set the rate to keep/drop all conditions as 0.3 and

Table 4: Ablation study on data distribution. We set the threshold between 'near' and 'far' to 50m. S: Satellite image P: Panoramas.

| $S$ | $P \geq 50m$ | $P < 50m$ | PSNR↑ | SSIM↑ | $P_{alex}$↓ | RMSE↓ | FID↓ | SD↑ |
|---|---|---|---|---|---|---|---|---|
| ✓ | | | 11.85 | 0.3143 | 0.5416 | 65.67 | 48.68 | 12.26 |
| | ✓ | | 12.20 | 0.3229 | 0.5244 | 63.10 | 42.81 | 12.37 |
| | | ✓ | 12.44 | 0.3224 | 0.5285 | 63.40 | 42.94 | 12.49 |
| ✓ | ✓ | | 12.67 | 0.3488 | 0.5174 | 57.12 | 39.71 | 12.61 |
| ✓ | | ✓ | 13.42 | 0.4136 | 0.5013 | 55.01 | 38.38 | 13.51 |
| ✓ | ✓ | ✓ | **14.14** | **0.4329** | **0.4343** | **51.42** | **33.68** | **13.60** |

Table 5: Ablation study. We analyze the effect of modality dropout and adding segmentation maps during the training stage.

| Ablation Objective | PSNR↑ | SSIM↑ | $P_{alex}$↓ | RMSE↓ | FID↓ | SD↑ |
|---|---|---|---|---|---|---|
| w/o modality dropout | 11.02 | 0.3215 | 0.5764 | 72.31 | 185.27 | 12.84 |
| w/o segmentation maps in training | 12.54 | 0.3665 | 0.5267 | 64.46 | 96.87 | 12.98 |
| Simple attention w/o geospatial attention adapter | 11.83 | 0.3275 | 0.5783 | 72.85 | 98.86 | 12.75 |
| Concat conditions and noise along spatial dimensions | 13.12 | 0.3589 | 0.5172 | 58.23 | 82.36 | 13.16 |
| Simple attention w/o geospatial information | 13.31 | 0.3896 | 0.4765 | 55.27 | 56.83 | 13.35 |
| Full | **14.14** | **0.4329** | **0.4343** | **51.42** | **33.68** | **13.60** |

0.1 respectively, and set the dropout rate of each condition to 0.1. For the text prompts, we randomly replace 50% of text prompts with empty strings to enhance the model's ability to learn image geometric relationships.

## 5.1 Quantitative Results

We compare the performance of our method against several baseline methods in Table 1 and cross-domain evaluation results in Table 2, using the Brooklyn and Queens dataset. We also show qualitative results in Figure. 6. For the baseline methods, Pix2pix Isola et al. (2017) is a traditional GAN-based image-to-image translation method. PanoGAN Wu et al. (2022) is a recent GAN-based cross-view synthesis method. Sat2Density Qian et al. (2023) is the state-of-the-art cross-view synthesis method.

As mentioned in previous works Zhu et al. (2021), in practical real-world applications, the panorama that users want to synthesis can occur at arbitrary locations in the area of interest, in this case perfectly aligned correspondence is not guaranteed. We both report the results on (1) the original Brooklyn and Queens dataset, in which satellite image and target location are not center-aligned and (2) our collected satellite images that are center-aligned with the target location. In both these two circumstances, our method outperforms all the cross-view synthesis methods. Specifically, our method shows more superiority on unaligned satellite data, while other cross-view synthesis methods have a significant performance degradation on unaligned satellite data. Our diffusion-based method has advantages especially in high-level metrics LPIPS and FID scores, showing its ability to generate high-fidelity images with global-level accuracy. Overall, our method shows more flexibility as it can achieve competitive performance without requiring the target panorama to be located at the center of the satellite image, improving the efficiency especially in parallelly generating multiple panoramas for a given region and bridging the gap between current research and practical applications.

## 5.2 Ablation Study

We conduct an ablation study that evaluates the impact of individual components of our proposed geospatial attention adapter on the resulting image quality. The results are shown in Table 3. Experimental results show that both the local and global geospatial attention can guide the diffusion model to better utilize the

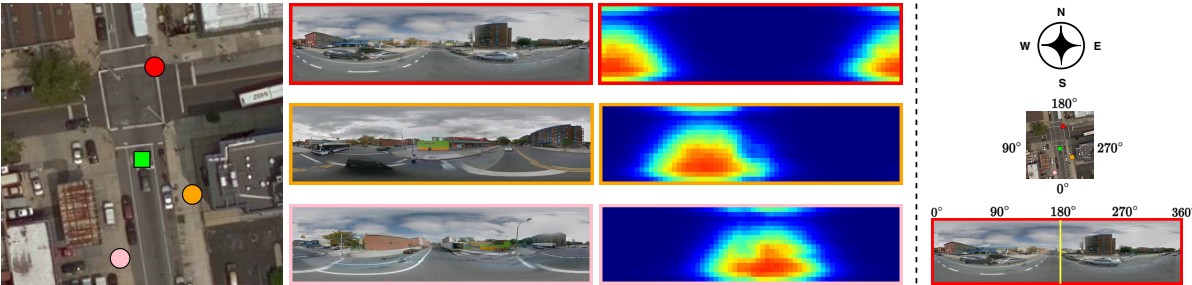

Figure 8: Visualization of local geospatial attention. The target location is represented by a green square in the satellite image. The nearby street-level panoramas (color-coded borders) are represented by same-colored circles in the satellite image.

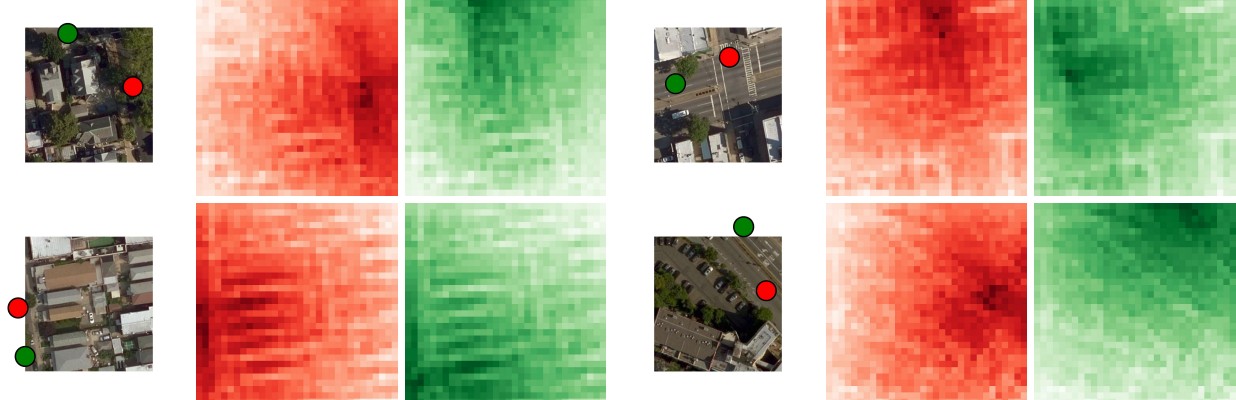

Figure 9: Visualization of global geospatial attention. The color-coded attention maps for two target locations are shown, corresponding to the same-colored dots in the satellite image. Darker colors represent more salient regions.

geometric relationships between mixed-view modalities. Furthermore, combining local and global geospatial attention helps supervise the model towards generating results with accurate layout distributions.

We also analyze the effect of the satellite images and near panoramas of different distances on our model. The results are shown in Table 4 and Figure. 7. In this experiment, we set thresholds to 50 meters based on the haversine distance between the near panorama and the target panorama to define 'near' and 'far'. We train our model on each subset and all available data. Panoramas that are near the target location have larger effect on the synthesis quality as there are more overlapping regions. However, the model does not rely on the 'near' panoramas to capture accurate geometry, since the model trained on 'far' panoramas and satellite images outperforms the model only trained on 'near' panoramas. The model trained on a subset of the data cannot synthesize accurate layout and sometimes confuses the sky region with trees. In summary, using all data significantly improves the synthesis quality of the model.

We conduct further ablation study in Table 5. We first conduct an experiment on modality dropout strategy. Compared with keeping all the input images, adding modality dropout achieves **24.6%↓** in $P_{alex}$ and **81.8%↓** in FID. We also remove the segmentation map branch of the multi-conditioned diffusion model in the training stage, which leads to severe performance degregation. Furthermore, we conduct experiments about: (1)concatenate conditions along the channel dimension and train a ControlNet to guide generation; (2) concatenate conditions and noise along spatial dimensions, followed by fine-tuning the diffusion model and (3) use simple attention to compute the attention map between the source view image and target view image, without adding location information and orientation information. All of these models show worse performance. These experiments demonstrate the effectiveness of modality dropout strategy and the supervision of segmentation maps in the training stage.

Table 6: Quantatitive comparison about Seamless.

|  | PanoGAN Wu et al. (2022) | Sat2Density Qian et al. (2023) | **Ours** |
|---|---|---|---|
| LRCE | 0.0897 | 0.0653 | **0.0436** |

### 5.3 Visualization of Geospatial Attention

We demonstrate the concept of geospatial attention visually. Figure. 8 shows an example of local geospatial attention, which uses the relative orientation and distance of a nearby street-level panorama in addition to semantic content. The street-level panoramas and paired local-level attention map use an east-north-up coordinate system, i.e., the center points to the north and the left/right boundary points to the south as Zhu et al. (2021). The target location is indicated by the green square in the satellite image and the location of the panoramas is indicated by the same-colored dot. Larger attention values (red) reflect the region in the panoramas that orient towards the target. Note that when the image and target locations are further apart, the high attention region shrinks, essentially reflecting the narrower field of view.

Figure. 9 shows an example of global geospatial attention for two target synthesis locations. Larger attention values (darker) reflect the regions in the overhead view that contribute the most. Note that, for the Brooklyn and Queens dataset, the street-level panoramas are primarily collected along the streets. This is captured in the global-level attention maps, which tend to show higher attention scores in the street-level regions near the target location (omitting buildings). Additional visualizations are shown in the supplementary material.

### 5.4 Discussion about Seamless

We adopt the Left-Right Consistency Error (LRCE) metric (Shen et al., 2022) to quantitatively assess the consistency along the left-right boundaries by analyzing the horizontal gradients. For depth evaluation, we integrate Depth Anywhere (Wang & Liu, 2024), a state-of-the-art panorama depth estimation method, to predict the depth maps of the panoramas. We compute the horizontal gradients of both the ground-truth depth maps and those of the generated panoramas, and then calculate the gradient differences. By randomly sampling 100 generated panoramas from the test set, we determined the mean gradient difference, as detailed in Table 6. Our experimental results indicate that, compared to previous cross-view synthesis methods, our approach achieves superior left-right boundary consistency, thereby demonstrating its seamless performance.

## 6 Conclusion and Future Work

We introduced the task of mixed-view synthesis, which extends the cross-view synthesis task to also include a set of nearby street-level panoramas as input. We proposed a novel multi-conditioned, end-to-end geospatial attention-guided diffusion framework for combining information from all input imagery to guide the diffusion-based synthesis process, achieving geometry-guided fine-grained spatial control. Unlike some cross-view approaches, our approach does not require a segmentation map of the target panorama during the inference stage or, like all previous cross-view approaches, that the target panorama be located at exactly the center of the satellite image. Together, these characteristics dramatically increase flexibility and ease of use. Experimental results demonstrate the effectiveness of our proposed model, in particular its ability to handle situations when the panoramas far from the target location.

For urban scenes, transient objects (e.g., cars, pedestrians) bring challenges to our method. Also due to the inherent randomness of the diffusion models, maintaining the view consistency between adjacent locations is still challenging in some detailed regions. Understanding how to reduce the influence of transient objects on guided diffusion models to synthesize clean and view-consistent panorama sequences would be a future research direction.

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
