# Appendix

## A  Dataset Details

The Brooklyn and Queens dataset that we adopt is an urban dataset with complex scenes. The original dataset contains overhead images downloaded from Bing Maps (zoom level 19) and street-view panoramas from Google Street View. We further collect the satellite images that are center-aligned with the panoramas, to enable a fair comparison with cross-view synthesis methods. Visualizations produced using our approach are based on a GeoDiffusion variant trained on the original dataset (not center-aligned). As the Brooklyn and Queens dataset consists of urban scenes with diverse buildings and transient objects (e.g., cars, pedestrians), it is more challenging for panorama synthesis than commonly used cross-view synthesis datasets like CVUSA Workman et al. (2015) and CVACT Liu & Li (2019), which consist of mostly rural scenes.

## B  Training Details

Our model is trained under batch size 16 deployed over 4 NVIDIA A100 80GB GPUs.

## C  Details about Geospatial Attention

During the local-level geospatial attention extraction, we first use a CNN encoder to extract features from the nearby street-level panoramas and the overhead image. The encoder consists of a ResNet-50 pretrained model and a 2D convolution with ReLU activation function. After that max-pooling and average pooling operations are applied along the channel dimensions. The shape of each panorama feature $F_i$ and the overhead image feature $S(l_t)$ are both $H \times W \times 2$.

For the distance information, from the input and target locations $(l_i, l_t)$, we calculate the geometric feature maps of the haversine distance $d$ between $l_i$ and $l_t$. The shape of the output distance feature is $H \times W \times 1$; For the orientation $\theta$ from the source to target panorama $l_t$, we compute it by rotating the original pixel rays, which are initially in the east-north-up coordinate frame, by computing the compass bearing between the source and target panorama, so that $[0, 1, 0]$ points to the target location. The shape of the orientation feature is $H \times W \times 3$.

The distance, orientation, the panorama feature map and the satellite image feature map are concatenated to a $H \times W \times 8$ feature tensor. Firstly, the feature tensor is passed through a $3 \times 3$ and a $5 \times 5$ convolution layer separately and concatenated to a temporal feature map, then the temporal feature map is passed to a $1 \times 1$ convolution layer with softmax activation function to get the spatial attention map $P_{i,t}$. After that the attention map $P_{i,t}$ is passed through an upsample layer to generate the local geospatial attention for the panorama image. The attention mask represents the geometric relationship between the nearby panorama and the target panorama.

For the global geospatial attention, the extracted local geospatial attention $P_{1,t}, P_{2,t}, \cdots, P_{i,t}$ are concatenated, and passed through the average pooling layers, the upsample module, the batch norm layer with sigmoid layer in sequence, and we get the global attention map. In the dataset, street-view near panoramas are collected along the streets, and in practical usage, the target locations are also mainly close to the streets. The global-level geospatial attention excludes the occlusions of buildings and guides the model to focus on the area around the target location with rich semantic information from the overhead view.

We also show further visualizations of geospatial attention in Figure. 2 and Figure. 3.

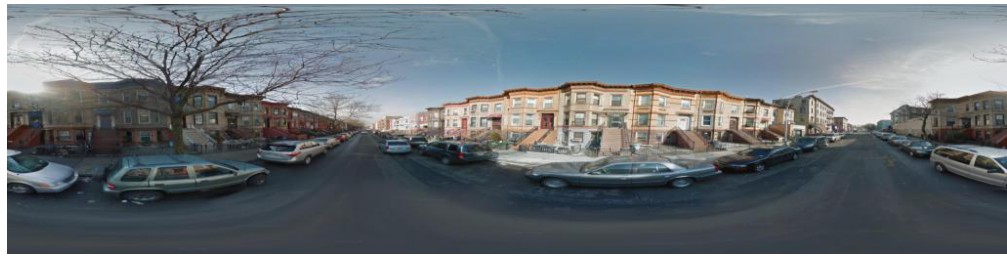

(a) Ground-truth

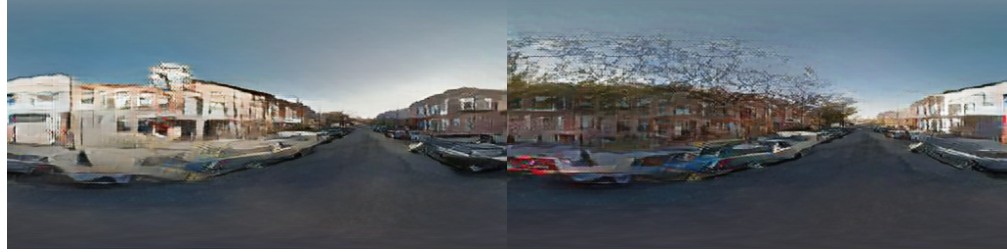

(b) Sat2Density-rotated

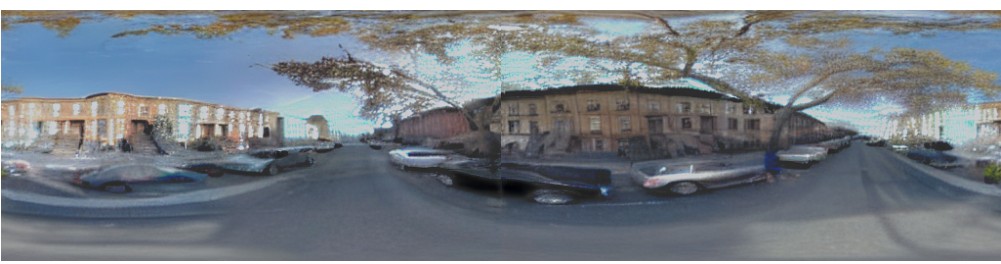

(c) Ours-rotated

Figure 1: Comparison about seamless of generated images.

## D    About Seamlessness

We follow the evaluation protocol established in prior cross-view synthesis works and evaluate the seamlessness of the generated panoramas, comparing the results versus previous state-of-the-art cross-view synthesis method, Sat2Density Qian et al. (2023). We randomly select one result and rotate it 180° about the vertical direction, which is shown in Figure. 1. The generated panoramas are not absolutely seamless, but compared with Sat2Density, our result is better in the concatenated area.

Note that different from works about panorama outpainting (Liao et al., 2024; Wu et al., 2023), we leverage the geospatial relationships among various input modalities to perform cross-view panorama generation, which existing panorama outpainting methods are not designed to accommodate.

## E    Additional Visualizations on Brooklyn

We show further visualizations on the Brooklyn subset in Figure. 4. Our model is trained on the original unaligned dataset. Compared with the cross-view synthesis methods that are trained on data with center-aligned satellite images, our GeoDiffusion method can synthesize accurate results both semantically and geometrically.

## F   Additional Visualizations on Queens

We show further cross-domain visualization results on Queens subset in Figure. 5. Compared with the cross-view synthesis methods that trained on data with center-aligned satellite images, our GeoDiffusion model can get accurate results with geometry accuracy. Results show the generalization ability of our proposed method.

## G   Camera Model

In the dataset, the satellite images approximate parallel projection and street-view panoramas follow spherical equirectangular projection. The panoramas in the Brooklyn and Queens dataset are with 360° horizontal and 180° vertical field of view, and are pre-rectified so that the center column line of the panorama represents the north direction.

## H   Discussions

In practical usage, we expect to collect street-view panoramas and satellite images ahead of time and preload them into the database. Our model is able to synthesize the panorama of a given location by using both the satellite images and the pre-loaded panoramas, creating a dense panorama field from the sparse panoramas. As our method reduce the reliance on aligned data, it is possible to allow synthesizing a bunch of locations in the satellite image region simultaneously, which improve the efficiency of creating dense panorama field.

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

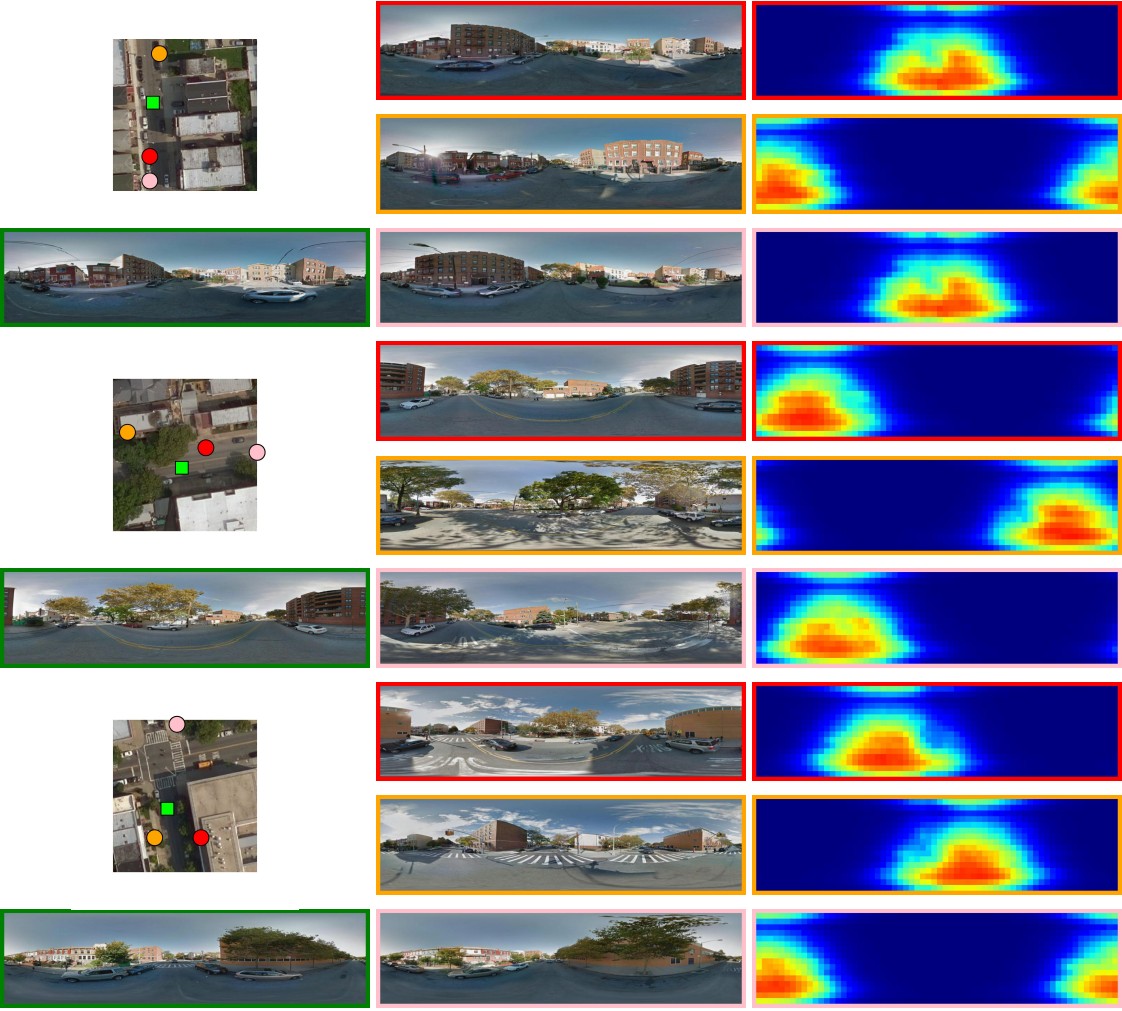

Figure 2: Additional visualizations of local geospatial attention. The target location is represented by a green square in the satellite image. The nearby street-level panoramas (color-coded borders) are represented by same-colored circles in the satellite image.

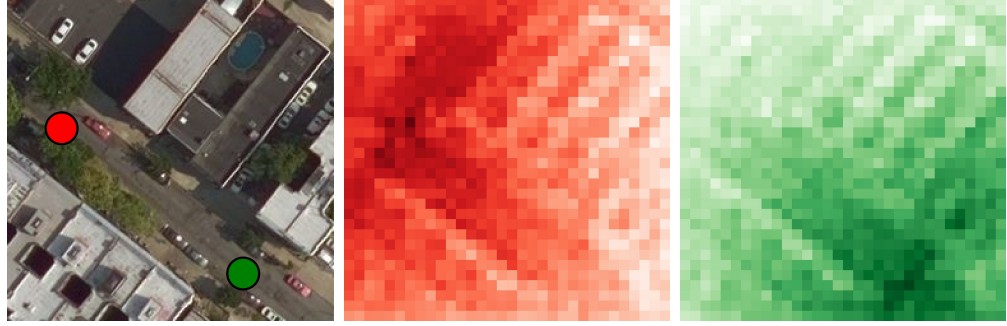

Figure 3: Visualization of global geospatial attention. The color-coded attention maps for two target locations are shown, corresponding to the same-colored dots in the satellite image. Darker colors represent more salient regions.

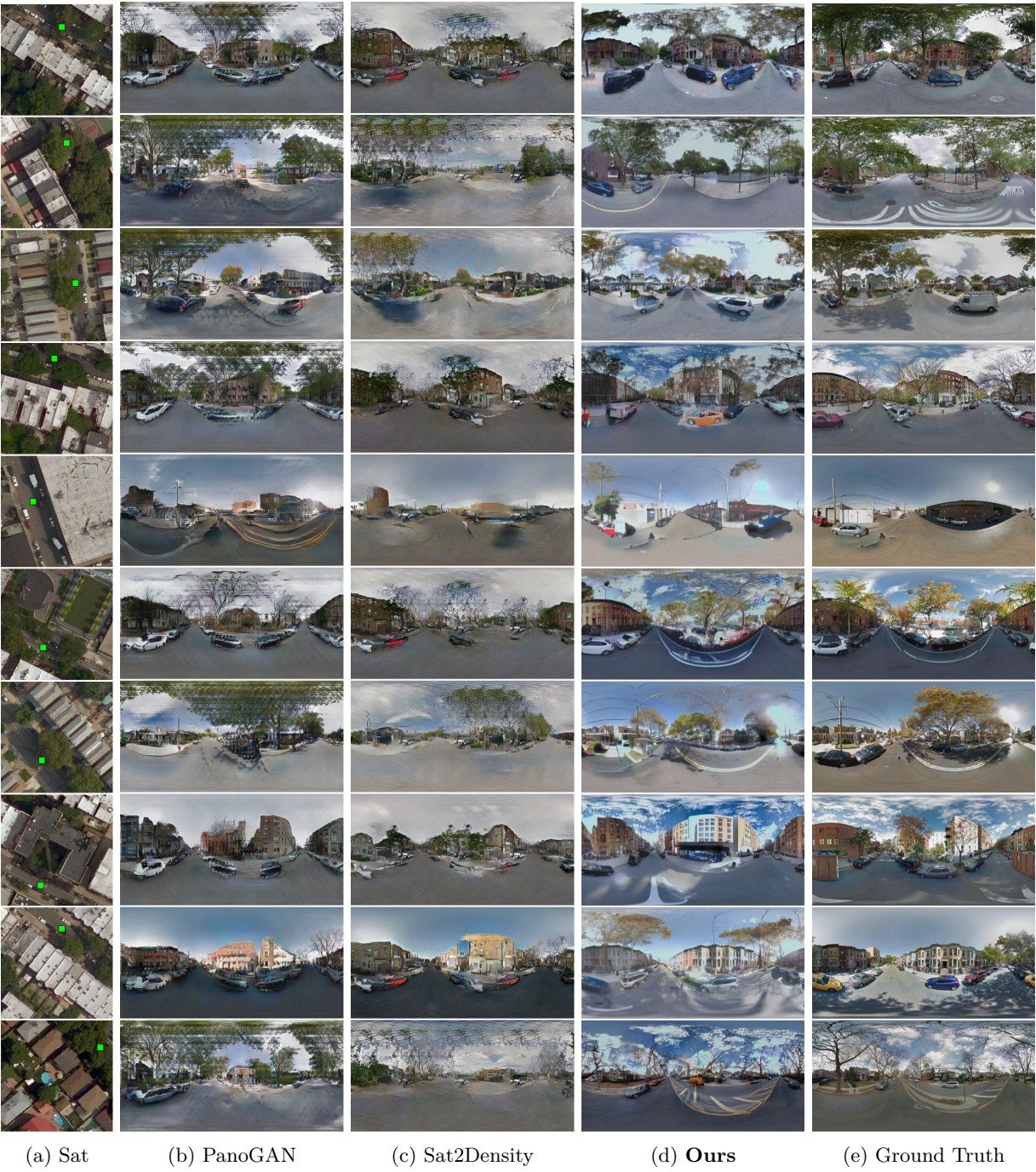

(a) Sat          (b) PanoGAN          (c) Sat2Density          (d) **Ours**          (e) Ground Truth

Figure 4: Additional qualitative results versus baselines on the Brooklyn test subset. The target location is represented by a green square in the satellite image. The cross-view synthesis methods that we compare with are trained on our collected center-aligned satellite images. Our approach, which integrates nearby street-level panoramas, is trained on the original satellite images (without center-aligned with the target location). Our method shows better results both geometrically and semantically.

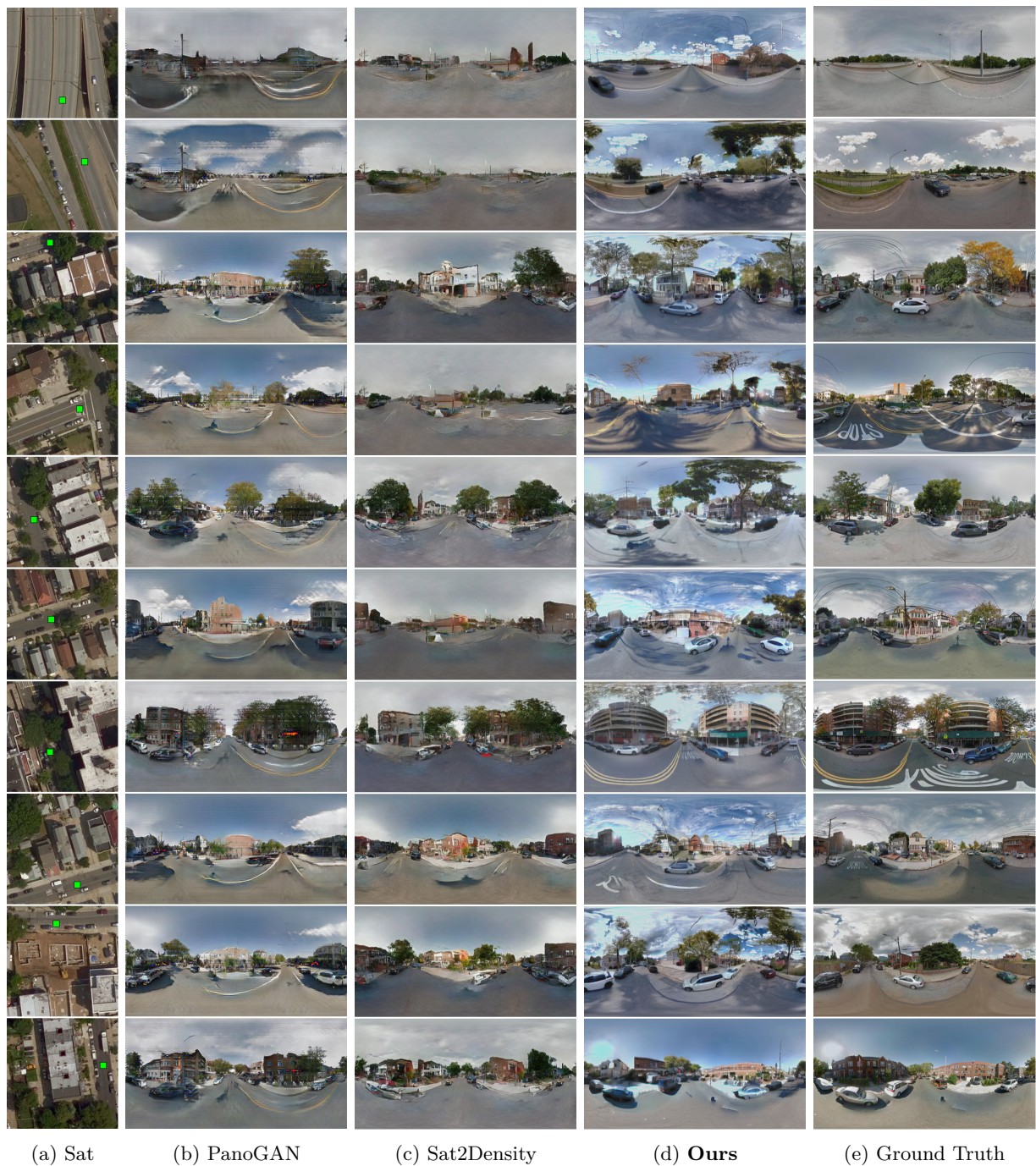

(a) Sat      (b) PanoGAN      (c) Sat2Density      (d) **Ours**      (e) Ground Truth

Figure 5: Cross-domain results versus baselines on the Queens subset. The target location is represented by a green square in the satellite image. The cross-view synthesis methods that we compare with are trained on our collected center-aligned satellite images. Our approach, which integrates nearby street-level panoramas, does not rely on the center-aligned satellite image. Our method not only generates more realistic results when compared to baselines, but more accurate results both semantically and geometrically when compared to the ground truth.