# OpenReview forum: "Mixed-View Panorama Synthesis using Geospatially Guided Diffusion"
_TMLR — Accepted by TMLR_

### Review · Reviewer_nsWF · 2025-02-17

**Summary Of Contributions:**

The authors propose a new task of synthesizing panorama images using mixed input signals. They also propose a diffusion-based approach via attention mechanisms as a potential solution to the proposed problem setting. The inputs are a combination of input panoramas and satellite imagery. The mixed use of inputs is important as not always we have dense panoramas for all points on earth, and only considering satellite imagery as inputs neglects the potential availability of sparse panoramas which is dense in information about textures and geometry of the area.

**Audience:**

Yes

**Claims And Evidence:**

No

**Requested Changes:**

The paper has strong motivation and thorough experiments. However, several areas need improvement. First, many sentences are too long, making it hard to follow. Shorter sentences would improve readability. Second, the naming of "global geospatial attention" is unclear and misleading, as it resembles transformer-based attention but is actually a differentiable weighting function. Third, some design choices, like using the Frobenius inner product and Hadamard product, lack proper justification or citations. Fourth, some references are inaccurate, such as citing Rombach et al. (2022) for conditional diffusion models. Fifth, the claim of introducing a novel multi-conditioned diffusion framework is exaggerated, as similar approaches exist in ControlNet variants. Lastly, the figures need better design, with less white space, consistent fonts, and uniform styles. Also, the Sat2Density results in Fig. 6 seem much worse than the proposed method, but Table 1 shows similar performance. Clarification on this discrepancy would be helpful.

**Strengths And Weaknesses:**

+ The work has a strong motivation for the proposed problem setting. Using satellite imagery to address the lack of dense panoramas for most regions, and using potential available panoramas for leveraging all the available data.
+ The experiments are thorough.

? In Fig 6, the Sat2Density results for the center-aligned scenario seem a lot worse that the proposed method. However, in Table 1, the results are almost the same between the proposed method and Sat2Density. Why is this the case?

- The writing could be significantly improved by cutting the long sentences into smaller ones. The paper is full of very long sentences which is hard for me as a reader to comprehend on the first pass. For example, this one: “To generate the target panorama Pt with precise layout distribution, the synthesis process should utilize the semantic information from P1, P2, · · · , Pn and S1, as well as the geometric relationships between street-level Pi → Pt, i ∈ (1, 2, · · · , n) and across satellite & street-level S1 → Pt.”. which spans almost 3 lines.
- The naming of the global geospatial attention is not the best in my opinion. I struggled to understand that whole section and had to read it back and forth multiple times, as I was expecting some sort of attention mechanism from the transformers literature. It however ended up being a differentiable weighting function.
- Certain design choices are not well-motivated. For example, the reason behind using the Frobenius inner product, or using the Hadamard product for fusion. These choices which are not necessarily widely used in the diffusion literature need to be backed with intuition or citations or thorough ablations.
- Some citations are not necessarily to the right papers. For example, consider this sentence: “Recently, conditional diffusion models Rombach et al. (2022) have become the state-of-the-art approach in the image synthesis task”. The paper Robach et al. (2022) is the paper introducing the latent-diffusion models, but wasn’t introducing conditional diffusion models.
- ControlNet variants with multiple control signals have been widely used in the literature. I believe it is a bit of stretch for the authors to claim they introduce “a novel multi-conditioned end-to-end diffusion framework”. The authors use the same architecture in the context of view synthesis from satellite and panorama imagery.
- The design of the figures could improve significantly. There is a lot of white space in the figures (e.g., Figure 4). The font family and font size of the figures don’t match the body of the paper. The font and style is not even consistent between different figures (notice Fig. 6 and Fig. 7). These are clear signs that more care could be put in the paper.

---

> ### Author Response · Authors · 2025-03-23
> **Response to Reviewer nsWF**
>
> Thanks a lot for your insightful feedback and kind advice.We will address your concerns one by one.
>
> 1. **About Writing:**
>
>     Thanks for your suggestions. We have made some changes in expression as suggested. As suggested we will change the expression “global geospatial attention” to “global weighting map” in the final version. Currently to prevent confusion of other reviewers, we do not change in the revised version, but in the camera ready version, we will do so.
>
> 2. **Mathematical Justifications:**
>
>     We would like to clarify the concepts of the Frobenius inner product and the Hadamard product.
>
>     - **Frobenius Inner Product:**
>
>         This binary operation takes two matrices of identical dimensions and returns a scalar, typically denoted as: $\langle\mathbf{A}, \mathbf{B}\rangle_{\mathrm{F}}$
>         It effectively computes the component-wise inner product of the matrices, treating them as vectors, and adheres to the standard axioms of an inner product.
>
>     - **Hadamard Product:**
>         Also known as the element-wise product, this binary operation takes two matrices of the same dimensions and returns a matrix of the multiplied corresponding elements. It is widely used in applying attention since it restricts the input feature following a weighted distribution.
>
> 3. **Citation Corrections:**
>
>     Thank you for highlighting the citation error. We have changed it into a different expression in the revised PDF.
>
> 4. **About Multi-Conditioned Diffusion Framework:**
>
>     Thanks for pointing out. We have already changed this part of expressions.
>
> 5. **Figure Formatting:**
>
>     The figure sizes have been adjusted in the updated version of the manuscript. We re-design some figures to better express the model design. We will do further revision in the final version after addressing other reviewers’ issues.
>
> 6. **About Fig.6 and Table 1**:
>
>     We acknowledge that low-level metrics may not fully capture human perceptual quality. High-level evaluation metrics such as FID and LPIPS are more indicative of human visual perception. As shown in Tables 1 and 2, our method significantly outperforms Sat2Density on these metrics, which means our method is visually more similar to the ground-truth image and reinforce the effectiveness of our approach.

---

> > ### Comment · Reviewer_nsWF · 2025-04-11
> > **Response to Rebuttal**
> >
> > Thank you for the thoughtful and detailed rebuttal. I appreciate the effort you put into addressing the points I raised. Most of my concerns have been adequately addressed, particularly regarding the writing clarity, terminology updates, and figure improvements. I'm also glad to see the clarifications around the mathematical operations and the correction of citations.

---

### Review · Reviewer_SqTH · 2025-02-24

**Summary Of Contributions:**

This work introduces the task of mixed-view panorama synthesis, which aims to synthesize a novel panorama given a small set of input panoramas and a satellite image of the area. It supports panorama synthesis for arbitrary locations worldwide. To achieve this, the authors propose a diffusion-based modeling and an attention-based architecture. Experimental results demonstrate the superiority of the proposed method.

**Audience:**

Yes

**Broader Impact Concerns:**

The method may be misused to generate deceptive or misleading geographic content. Biases or inaccuracies in AI-generated panoramas could impact urban planning and environmental assessments. Privacy concerns also arise if sensitive geospatial imagery is used without authorization.

**Claims And Evidence:**

Yes

**Requested Changes:**

Please refer to the above Weaknesses for more details.

**Strengths And Weaknesses:**

Strengths:

+ The task of mixed-view panorama synthesis that uses a satellite image and a sparse set of nearby panoramas looks interesting. It also shows the potential to apply to diverse scenes.
+ A geospatial attention as the geometry constraint is proposed to achieve geometry-guided fine-grained spatial control.
+ The paper is easy to follow, and the organization is well structured.

Weaknesses:

- While the proposed new task looks interesting and practical, the technical contributions presented in this work seem to be somehow weak. For example, the proposed local attention and global attention extractions are similar to normal attention operations, and some of them are derived from previous works. Besides, the proposed multi-conditioned diffusion model also shares similar processes with ControlNet. The authors are suggested to highlight the technical contribution of this work.
- This work targets panorama synthesis; however, the specific properties of panoramic images have been ignored. For instance, the panorama shows dramatic geometric distortions compared to the perspective images. How did the authors ensure the generated results kept this property? It might be improper to evaluate the generated panoramas only using the metrics designed for perspective images, such as PSNR and SSIM. Could we propose a new metric to specifically evaluate this geometric distortion? I believe some generated panoramas would obey the accurate geometric distortion distribution while looking reasonably at the appearance level.
- In addition to the above concern, the property of seamless content in generated panoramas has also been omitted. Namely, the panorama can be stitched from left to right boundaries without a noticeable edge effect. Did the authors leverage the cylinder convolutions like previous panoramic vision works? For example, "Cylin-Painting: Seamless 360° Panoramic Image Outpainting and Beyond" and "PanoDiffusion: 360-degree Panorama Outpainting via Diffusion" use a cylinder-like convolution or circular padding to ensure the seamless content generation of panoramas. The brief reviews and discussions of these works are also expected to be presented in this work.
- Some previous works propose specific metrics to evaluate the panorama model. For example, "PanoFormer: Panorama Transformer for Indoor 360 Depth Estimation" proposes the Left-Right Consistency Error (LRCE) to measure the consistency of the left-right boundaries by calculating the horizontal gradient. The experiments should add this metric for generated panorama evaluations.

---

> ### Author Response · Authors · 2025-03-23
> **Response to Reviewer SqTH**
>
> Thanks a lot for your insightful feedback and kind advice. We will address your concerns one by one.
>
> 1. **About Technical Contributions:**
>
>     There are no previous works that use geospatial relationships as constraints in image/panorama synthesis for urban scenes. Our work proposes a geospatial attention-based unified diffusion framework, which uses geospatial attention as the geometric constraint to associate different source views with the target view, achieving geometry-guided spatial control. Our framework is the first to use geospatial relationships as a constraint in outdoor panorama synthesis tasks.
>
> 2. **About Geometric Distortion:**
>
>     Thanks for your suggestions. To better evaluate the geometric distortions, we propose a Center Root Mean Squared Error(C-RMSE) to calculate the weighted RMSE score. As the boundary regions of the panorama contains more distortions, we reduce the weight of these pixels to 0.2 when calculating the metrics. In the experiment we set the boundary width to 20 pixels. The results on Brooklyn test set are shown in the table below, following the center-aligned setting. Compared with PanoGAN and Sat2Density, our method still get better performance in these distortion-aware evaluation metrics, which demonstrate the effectiveness of the method.
>
> Table A: Results about geometric distortion
>
> |       | PanoGAN | Sat2Density | Ours   |
> |-------|--------|-------------|--------|
> | C-RMSE | 53.25  | 48.04       | 45.62  |
>
> 3. **Visualization:**
>
>     Note that for visualization, we have already provided qualitative results in the supplementary Figure 1 compared with Sat2Density. In the comparison, we use one result in Fig. 6 and rotate it 180-degree about the vertical direction, which shows that our method is more seamless in the concatenated area. The generated panoramas are not absolutely seamless, but other methods also have the same issue. Compared with Sat2Density, our result is better visually in the concatenated area.
>
>     Our work is different from inpainting or outpainting-based panorama generation methods. As it is based on diffusion models without convolution, it is hard to directly use cylinder-like convolution and circular padding in our model. However, as suggested, we have already discussed the seamlessness of the panorama and illustrated the differences between our work and Cylin-Painting[1] and PanoDiffusion[2] in the revised version, and we have cited these papers in section 5.4(marked red) in the revised version. Feel free to have a look.
>
> 4. **About Metrics for Panorama Model:**
>
>     As suggested, we adopt the Left-Right Consistency Error (LRCE) metric[4] to quantitatively assess the consistency along the left-right boundaries by analyzing the horizontal gradients. For depth evaluation, we integrate Depth Anywhere[3], a state-of-the-art panorama depth estimation method, to predict the depth maps of the panoramas. Since in PanoFormer[4]they do not release their metrics evaluation code, we reimplemented the metric by computing the horizontal gradients of both the ground-truth depth maps and those of the generated panoramas, and then calculating the gradient differences. By randomly sampling 100 generated panoramas from the test set, we determined the mean gradient difference, as detailed in the table below. Our experimental results indicate that, compared to previous cross-view synthesis methods, our approach achieves superior left-right boundary consistency, thereby demonstrating its seamless performance. We have added section 5.4(masked red) to discuss this in the revised PDF. We have also already cited this paper in the this part.
>
>
> Table B: Quantitative comparison about Seamless.
> |      | PanoGAN | Sat2Density | Ours   |
> |------|--------|-------------|--------|
> | LRCE | 0.0897 | 0.0653      | 0.0436 |
>
>
> [1] Cylin-Painting: Seamless 360° Panoramic Image Outpainting and Beyond. TIP 2024
>
> [2] PanoDiffusion: Depth-aided 360-degree Indoor RGB Panorama Outpainting via Latent Diffusion Model, ICLR 2024
>
> [3] Depth Anywhere: Enhancing 360 Monocular Depth Estimation via Perspective Distillation and Unlabeled Data Augmentation, NeurIPS 2024
>
> [4] PanoFormer: Panorama Transformer for Indoor 360 Depth Estimation. ECCV 2022

---

> > ### Comment · Reviewer_SqTH · 2025-04-10
> > **Reponse to rebuttal**
> >
> > Thanks for your detailed rebuttal, which well addressed most of my concerns.

---

### Review · Reviewer_qnV2 · 2025-03-18

**Summary Of Contributions:**

This paper tackles the problem of generating a new panorama using both a satellite image and a few input panoramas. Unlike previous work that uses only one of these inputs, the method combines both sources of information.
The authors adopt a ControlNet-style module to incorporate all the input conditions into the generation process.
They introduce a complex attention extraction module designed to balance information from different geolocations accurately.

**Audience:**

Yes

**Claims And Evidence:**

No

**Requested Changes:**

1. Add ablation studies comparing other ways of injecting conditions into the model to validate the proposed method. Suggested baselines include:
- Concatenating conditions along the channel dimension and training a ControlNet to guide generation.
- Concatenating conditions and noise along spatial dimensions, followed by fine-tuning the diffusion model.
- Using simpler architectures, such as pure convolution or pure attention layers, to handle the conditioning.
2. How is the segmentation map used during training? It is not shown in Figure 3 or Figure 5. Given its importance (as suggested by Table 5), why doesn’t this create a training–inference mismatch?

3. Include comparisons with other diffusion-based models to better demonstrate the advantages of your approach.

4. In Figure 2, “extended prompts” appear in both the Geospatial Attention Extractor and the diffusion model. However, the paper does not explain what they are or how they are used. Please add a detailed explanation.

5. Why does the copied encoder also take noise as input? Please explain the reasoning behind this design.

**Strengths And Weaknesses:**

Strengths:

- The proposed method effectively combines multiple input conditions to generate consistent target panoramas.
- The geospatial attention module appears to successfully learn and capture accurate spatial relationships across locations.

Weaknesses:
- The framework is overly complex, and many modules are not well-explained or justified. There’s no clear evidence showing that these design choices outperform simpler alternatives.
- Key ablation studies and comparisons with competitive baselines are missing.
- The paper does not clearly demonstrate the advantages of mixed-view panorama synthesis compared to using only single-condition inputs.

---

> ### Author Response · Authors · 2025-03-23
> **Response to Reviewer qnV2**
>
> Thanks a lot for your insightful feedback and kind advice. We will address your concerns one by one.
>
> **About the advantages of mixed-view panorama synthesis:**
>
> For the advantages compared with using single-condition input, we have experiments in Table 4 about this. Compared with only using single modality, our method get better performance, which demonstrates the effectiveness of combining both modalities in synthesizing the panorama of the target location.
>
> **About your suggested changes:**
>
> 1. **About further ablation study:**
>
> We rely on the pretrained model of ControlNet, which is based on the Stable Diffusion V1.5 pretrained weights. For the points (1) and (2), if we change the channel or spatial dimensions of the input, we cannot use the pretrained model for initialization, which would lead to a severe drop in generation quality. Moreover, these modifications represent a fundamentally different model design rather than a variant suitable for ablation studies.
>
> For point (3), we add experiment that use simple attention to compute the attention map between the source view image and target view image, without adding location information and orientation information. We use convolution to extract image features and calculate the QKV attention score between the images features. The results are shown in the Table A below.  Such ablation study shows the effectiveness of our method in utilizing geospatial information in outdoor panorama synthesis. We have also updated this in the revised PDF(masked red).
>
> Table A: Further ablation study.
> |                                      | PSNR↑ | SSIM↑  | P_alex↓ | RMSE↓ | FID↓  | SD↑   |
> |--------------------------------------|-------|--------|---------|-------|-------|-------|
> | simple attention w/o geospatial information | 13.31 | 0.3896 | 0.4765  | 55.27 | 56.83 | 13.35 |
> | Ours                                 | 14.14 | 0.4329 | 0.4343  | 51.42 | 33.68 | 13.60 |
>
> 2. **Segmentation Map Input:**
>
> Our architecture incorporates a segmentation map input through an independent ControlNet branch. This branch uses segmentation maps solely as a control signal (analogous to the original ControlNet design) and is not involved in geospatial attention extraction or fusion. We have updated Figure 2 in the revised PDF to clarify this architecture. Training with segmentation maps of the target location not only accelerates convergence but also enables the model to capture semantic cues effectively. Consequently, during inference, the model operates without relying on these inputs, which aligns with practical application scenarios.
>
> 3. **Comparison with Diffusion-based methods:**
>
> Previously there is no work about diffusion-based methods in cross-view synthesis. We have already compared our result with methods that use only diffusion as the backbone for cross-view synthesis in Table 4 row 1, which only take the branch of satellite image during training.
>
> 4. **About Prompt:**
>
> As mentioned in the paper, we use "A high-resolution street-view panorama" as the default prompt. To prevent any misunderstanding, we have adjusted the figure 2 and remove the term "extended" in the revised PDF.
>
>
> 5. **About copied encoder:**
>
> Our overall architecture remains consistent with the original ControlNet design. Each copied encoder branch operates as an independent diffusion process, just like the original diffusion branch. In ControlNet, as introduced in the original paper, the “copied encoder” refers to a branch of the U‑Net that is structurally identical to the original U‑Net encoder but specializes in handling the extra conditioning input (e.g., edge maps, segmentation masks, etc.).  Our method is based on ControlNet, which, in each copied encoder branch of condition, use the same denoise process as the original stable diffusion.

---

> > ### Comment · Reviewer_qnV2 · 2025-03-31
> > **Response**
> >
> > "For the points (1) and (2), if we change the channel or spatial dimensions of the input, we cannot use the pretrained model for initialization, which would lead to a severe drop in generation quality. "
> >
> > No, these two baselines are well-established methods for integrating additional conditions into diffusion models. Baseline 1 modifies only the inputs to the ControlNet, while Baseline 2 can effectively incorporates conditions by fine-tuning only the text-conditioning component of the diffusion model.

---

> ### Author Response · Authors · 2025-04-02
> **Response to reviewer qnV2**
>
> Thanks for your response.
>
> As we mentioned in page 7 section 4.2, in our framework, the conditions are concatenated in the channel dimension, passed through the geospatial attention adapter, and then the attention-guided features for each condition are injected into
> the corresponding copied encoder, with noise added according to the time embedding. For the point(1) you mentioned, we think it is exactly just moving all the geospatial attention adapter. So we conduct experiments about this. The result is shown below, which shows the effectiveness of our model design.
>
>
> Table B: Further ablation study.
> |                                      | PSNR↑ | SSIM↑  | P_alex↓ | RMSE↓ | FID↓  | SD↑   |
> |--------------------------------------|-------|--------|---------|-------|-------|-------|
> | simple attention w/o geospatial attention adapter | 11.83 | 0.3275 | 0.5783  | 72.85 | 98.86 | 12.75 |
> | Ours                                 | 14.14 | 0.4329 | 0.4343  | 51.42 | 33.68 | 13.60 |
>
>
> For point(2) you mentioned, we are still not sure what do you mean by spatial dimension. Note that our pipeline focus on the utilizing the geospatial relationships between source location image and the target location. We do not rely on fine-tuning text-condition to fine-tune the diffusion model since we give a general prompt to all the input conditions, and we just rely the original text-to-image ability of the diffusion models. Our previous ablation studies have already confirmed that our model can effectively capture and utilize geospatial relationships during the image generation process.

---

> > ### Comment · Reviewer_qnV2 · 2025-04-03
> >
> > Baseline 2 Implementation: Simply concatenate the input image (noise) of size H×W×C with the condition of size HxWxC, forming an input of H×(2W)×C. Input this into the diffusion model so it can leverage its well-trained attention layers to capture spatial relationships effectively.

---

> > > ### Author Response · Authors · 2025-04-07
> > > **Response to Reviewer qnV2**
> > >
> > > Thanks for your detailed explanation. As you suggested, we implemented the baseline 2 through concatenating the input image and noise, then input this into the diffusion model. The results is shown below. From quantitative comparisons, we can see that the generated quality is still worse than our proposed framework. The diffusion model cannot learn to capture such complex geospatial relationships well by the baseline 2 method.
> > >
> > > Table C: Further ablation study on baseline 2.
> > > |                                      | PSNR↑ | SSIM↑  | P_alex↓ | RMSE↓ | FID↓  | SD↑   |
> > > |--------------------------------------|-------|--------|---------|-------|-------|-------|
> > > | Concatenating conditions and noise along spatial dimensions | 13.12 | 0.3589 | 0.5172  | 58.23 | 82.36 | 13.16 |
> > > | Ours                                 | 14.14 | 0.4329 | 0.4343  | 51.42 | 33.68 | 13.60 |
> > >
> > >
> > > Please feel free to let us know if you have any remaining questions or if further clarification is needed. We are happy to discuss.

---

> > > > ### Comment · Reviewer_qnV2 · 2025-04-07
> > > >
> > > > Thank you for your response. A better way to implement this baseline could be using the inpainting version of the diffusion model, but the current version works fine for a controlled ablation study. I don't have any other issues. I strongly recommend adding these experiments and analysis in your revision.

---

> > > > > ### Author Response · Authors · 2025-04-07
> > > > > **Response to Reviewer qnV2**
> > > > >
> > > > > Thanks for your suggestions. I have already added these experiments and analysis in the revised PDF.

---

### Decision · Action_Editor_x5UM · 2025-04-28

**Recommendation:** Accept with minor revision

**Comment:**

The reviewers all agreed that the paper warranted publication.  A number of minor issues around presentation and additional ablations arose during the discussion which should be incorporated into the final paper.

**Audience:**

The problem is a variation on conditional generation and is solved with diffusion models.  It will be of interest to at least a reasonable portion of the TMLR audience.

**Claims And Evidence:**

The claims in the paper are well supported by experimental evidence.